# Validation of an Automatic Inertial Sensor-Based Methodology for Detailed Barbell Velocity Monitoring during Maximal Paralympic Bench Press

**DOI:** 10.3390/s22249904

**Published:** 2022-12-16

**Authors:** Lorenzo Rum, Tommaso Sciarra, Nicoletta Balletti, Aldo Lazich, Elena Bergamini

**Affiliations:** 1Department of Movement, Human and Health Sciences, University of Rome “Foro Italico”, Piazza L. De Bosis 6, 00135 Rome, Italy; 2Defense Veterans Center, Celio Army Medical Center, 00184 Rome, Italy; 3Department of Biosciences and Territory, University of Molise, 86100 Campobasso, Italy; 4DIAG, Sapienza University of Rome, 00185 Roma, Italy

**Keywords:** Paralympic powerlifting, bench press, barbell velocity, inertial sensor, strength training

## Abstract

Current technologies based on inertial measurement units (IMUs) are considered valid and reliable tools for monitoring barbell velocity in strength training. However, the extracted outcomes are often limited to a few velocity metrics, such as mean or maximal velocity. This study aimed at validating a single IMU-based methodology to automatically obtain the barbell velocity full profile as well as key performance metrics during maximal Paralympic bench press. Seven Paralympic powerlifters (age: 30.5 ± 4.3 years, sitting height: 71.6 ± 6.8 cm, body mass: 72.5 ± 16.4 kg, one-repetition maximum: 148.4 ± 38.6 kg) performed four attempts of maximal Paralympic bench press. The barbell velocity profile and relevant metrics were automatically obtained from IMU linear acceleration through a custom-made algorithm and validated against a video-based reference system. The mean difference between devices was 0.00 ± 0.04 m·s^−1^ with low limits of agreement (<0.09 m·s^−1^) and moderate-to-good reliability (ICC: 0.55–0.90). Linear regression analysis showed large-to-very large associations between paired measurements (*r*: 0.57–0.91, *p* < 0.003; SEE: 0.02–0.06 m·s^−1^). The analysis of velocity curves showed a high spatial similarity and small differences between devices. The proposed methodology provided a good level of agreement, making it suitable for different applications in barbell velocity monitoring during maximal Paralympic bench press.

## 1. Introduction

In the last decade, assessing the velocity of movement execution has become a key aspect to evaluate the quality of a session in several strength training methodologies, such as velocity-based training [1,2,3]. Among the various strength exercises, those involving the use of a barbell (e.g., squat, deadlift and bench press) have been the most investigated by the scientific community [4,5,6,7,8,9]. Recent advances in technology as well as reduced costs and higher accessibility have led IMUs to succeed as highly adopted solutions in sport practice. Evidence exists that IMUs can be considered valid and reliable tools to estimate barbell velocity trajectories during different barbell exercises [10]. In their review, Clemente and colleagues reported the level of validity and reliability of eight different IMU models which are currently available in the market and adopted in the assessment of barbell kinematics by both scientific and sports communities [10]. Most of the commercially available models are easy to use as they only require the fixation of the sensor on the barbell. Furthermore, they can provide almost instantaneous feedback to the user via a smartphone interface (i.e., mobile app) released by the manufacturer, which makes them a user-friendly solution to monitor training load. Despite the continuous data sampling at high-frequency rates (up to ~1000 Hz), the parameters extracted from the linear acceleration/velocity profiles are often discrete (i.e., mean, or maximum velocity) and mainly computed over the pushing/pulling phase of the exercise (i.e., concentric phase). As a consequence, current IMU-based methodologies have failed in providing more subtle information about the different phases and specific events of barbell exercises. Providing such a comprehensive description of the movement would help in uncovering key aspects of motor action, thereby being of great benefit to sports specialists and practitioners for improving their performance.

The Paralympic powerlifting discipline has seen an increase in participation and competition achievements at the international level, being one of the fastest-growing sports in the Paralympic movement [11,12,13]. As an adapted version of powerlifting, it only involves one single exercise, that is, a maximal bench press lift (one repetition maximum, 1RM) on a flat bench. This version of the bench press exercise highly limits the engagement of the athlete’s lower body as both legs are extended and secured with straps on the bench during execution [14]. The lifting sequence of the Paralympic bench press is very close to the classic version of the exercise and can be divided into two main phases. First, the barbell is lowered from a starting position with both elbows extended up to the athlete’s chest during the eccentric phase. Here, the barbell velocity is negative as it is accelerating towards the floor, showing an inverted bell-shaped profile with zero-velocity tails (i.e., before the initial lowering of the barbell and at the stop at the chest). Second, in the concentric phase, the barbell is lifted as symmetrically as possible until both elbows are completely extended. The barbell velocity profile during this lifting phase is characterized by a rather repeatable shape, particularly at higher load intensities (>80% 1RM) [15]. After the stop at the chest, the barbell is accelerated upwards until reaching the so-called sticking point, when deceleration is observed. The velocity curve is characterized by a positive peak at the sticking point and the athlete’s capacity to overcome this point is fundamental for the successful execution of the lift [15]. After the sticking point, a second positive peak of the velocity curve occurs as the barbell is accelerated and then decelerated before the complete extension of both elbows. The correct technical execution of this lifting sequence is assessed by referees to validate the lift performance during competition. Therefore, accurately assessing the specific features of the barbell velocity curve during the lifting sequence is crucial for performance evaluation and improvement.

To date, both linear position transducers [16,17,18,19] and video-based solutions [15,20,21] have been used to assess barbell velocity in Paralympic bench press, albeit with some drawbacks. On one hand, linear position transducers are valid solutions to monitor barbell velocity [22,23,24] but they do only provide general information about the lifting (concentric) phase of the movement. Furthermore, they require time and precision for device positioning which limits their use in daily sports practice. On the other hand, video-based solutions allow to evaluate the various features of the barbell velocity profile [15,20,21] but they do often require space for optimal camera positioning and time-consuming post-processing with dedicated software for video analysis (e.g., Kinovea). Therefore, from an applicability standpoint, there is a need for new solutions able to provide both event-specific and time-efficient accurate measurements of the barbell velocity.

Recently, an IMU-based algorithm to automatically identify the specific events of the Paralympic bench press has been proposed and validated [25]. The algorithm showed a high performance in event detection (detection error rate 2.8%) with an average level of error across events below 0.12 s (~3% total task duration) and moderate-to-excellent reliability. These results suggested that this approach may be implemented to estimate the velocity parameters specific to the main phases of the maximal Paralympic bench press. The aim of the present study was therefore to validate a single IMU-based methodology to obtain barbell velocity profile and metrics at key time events of maximal Paralympic bench press. This novel approach, consisting in the combination of the automatic event detection algorithm with IMU technology, will provide coaches and athletes with a more detailed picture of the bench press lifting technique, thus supporting the identification of factors limiting performance and in the definition of personalized training protocols. Specifically, a more detailed description of the barbell velocity profile and performance-related metrics will be provided compared to the existing methods. In addition, the proposed methodology will be deeply validated and specific statistical tests will be performed to compare discrete parameters and the similarity of velocity profiles with a video-based measurement system.

## 2. Materials and Methods

### 2.1. Participants

Seven Paralympian powerlifters (1 female, 6 males) of elite international level from the Italian national team participated in the study. Participants were included if they were affected by at least one eligible impairment for Paralympic powerlifting competition [26] and they had more than 5 years of experience in strength or powerlifting training. Demographic data and types of impairment of the sample are presented in Table 1. All participants underwent a classification procedure and were declared eligible to participate in Paralympic competitions according to international regulation [26]. Participants were excluded if impairment or injury in the upper limbs was reported on the day of testing. The study protocol was approved by the institutional review board (University of Rome “Foro Italico”: CAR 116/2022) and written informed consent was provided by all participants before testing.

### 2.2. Data Collection

Data collection was performed during a mid-season retreat. A high-resolution and high-speed camera (Hero 9, GoPro, San Mateo, CA, USA) was used as a reference device to estimate barbell trajectory during the exercise. The camera was positioned on a tripod at an approximate height of 1 m from the floor and perpendicular to the barbell longitudinal axis to record the 2D position of a reflective marker (diameter 17 mm) that was placed laterally on the barbell endcap (Figure 1). The video was captured at a sampling frequency of 120 Hz and the open-source software Kinovea (version 0.9.5) was used to obtain the vertical displacement of the marker [27]. The diameter of the barbell sleeve at the picture frame immediately before the exercise execution was used for the software calibration to minimize the depth error of the sampled data. To further improve the accuracy of the marker tracking by the software, a zoom of 600% was applied for the frame-by-frame visual inspection and a manual correction was applied when needed. A single IMU (MTw, Xsens, Enschede, The Netherlands) was used to measure the barbell kinematics at a sampling rate of 100 Hz and was time-synchronized with the video-based system. The unit was fixed on the barbell on the same side as the reflective marker and close to the handgrip without interfering with the athlete’s hand positioning. Data from the 3D linear accelerometer of the IMU (full range scale ±160 m·s^−2^) was used for the analysis. A six-position static test was performed before testing to calibrate accelerometer data [28]. This calibration procedure allows to extract the estimates of accelerometer axial bias and sensitivity scaling factor which were then used to minimize the error due to sensor measurement miscalibration, as follows:X_cal_ = (X_raw_ − b) × s × g,(1)
where X_cal_, X_raw_, b, s and g represent, respectively, the calibrated axial accelerometer data (m·s^−2^), the raw axial accelerometer data (m·s^−2^), bias (m·s^−2^), sensitivity scaling factor (unitless) and the gravitational acceleration (m·s^−2^).

The testing procedure consisted of a standardized warmup (Table 2) followed by four attempts of 1RM Paralympic bench press [29]. The standardized warmup was based on subjects’ most recent result of 1RM in competition and consisted in 20 repetitions with unloaded barbell (20 kg), 2 sets of 6 repetitions at 40% 1RM, 1 set of 3 repetitions at 60% 1RM, 1 set of 2 repetitions at 75% and 1 set of 1 repetition at 85% 1RM, with a 3-min pause between sets [29]. Warmup sets were performed with no indication for execution speed and three-minute rest pauses were provided between the sets. Considering the maximal nature of the exercise, assistance was provided by experienced technical personnel at each lift to prevent the risk of injury in case of failure. To limit the effect of fatigue, rest pauses of five minutes were provided at the end of the warmup and after each attempt, with the athlete’s recovery being monitored with the total quality recovery scale before proceeding with the next attempt [30]. Tests were performed using an official Paralympic powerlifting bench and barbell (Eleiko, Halmstad, Sweden), approved by the International Paralympic Committee. During the lift, a system of belts was used to secure the lower body of the athlete to the bench according to the World Para Powerlifting technical regulations [14]. Data from the valid lifts (i.e., lifted with the correct technique validated by a technical officer according to [14]) were considered for the analysis.

### 2.3. Data Analysis

The barbell velocity profile was obtained from the two measurement systems as follows. The vertical displacement of the marker acquired through video recordings was low-pass filtered by a 2nd order Butterworth filter with a cut-off frequency of 20 Hz and then the first derivative was calculated to obtain the barbell velocity [21]. As regards the IMU data processing, the norm of the tri-axial linear acceleration was calculated after the calibration and gravitational acceleration was removed. The obtained signal was then low-pass filtered by a 2nd order Butterworth filter with a cut-off frequency of 10 Hz and numerically integrated to obtain the barbell velocity.

The timings of specific events of the Paralympic bench press movement were automatically identified from IMU- and video-derived data through a previously validated algorithm [25]. Briefly, this custom-made algorithm exploited the in-built functions of Matlab^®^ software (version 9.8, Mathworks, Natick, MA, USA) to automatically detect the events based on the identification of peaks, slopes or thresholds on the signal variability (please refer to [25] for detailed information on the algorithm). The following six events were detected (Figure 2): the initial bar lowering (Start), the stop of the barbell on the chest (Stop), the first peak in velocity (i.e., the sticking point, Vmax1), the minimum decrease in velocity (Vmin) and the last velocity peak (Vmax2) before the complete elbow extension (End). Discrete velocity parameters from video and IMU data were then obtained at Vmax1, Vmin and Vmax2 events, and within the following time intervals:Between Start and Stop (eccentric mean/maximum velocity: EmV/EMV);Between Stop and End (concentric mean/maximum velocity: CmV/CMV);Between Stop and Vmax1 (pre-sticking region mean velocity: PreStickV);Between Vmax1 and Vmin (sticking region mean velocity: StickV);Between Vmin and Vmax2 (post-sticking region mean velocity: PostStickV).

For the evaluation of curve similarity between the two measurement systems, the time-series data of barbell velocity were also segmented from Start to End events and time-normalized to 100 points. Data processing and analysis were performed using MATLAB^®^ software (R2020a, MathWorks Inc., Natick, MA, USA).

### 2.4. Statistical Analysis

#### 2.4.1. Discrete Velocity Parameters

The assumptions of data normal distribution and homoscedasticity in discrete velocity parameters were verified using the Kolmogorov–Smirnoff test and Brown–Forsythe test, respectively. For all discrete velocity parameters, the level of agreement between the IMU- and video-based measurement systems was evaluated through the Bland–Altman analysis, which indicates the systematic bias and the 95% limits of agreement (LoA) [31]. The validation analysis also included the following set of statistical tests to evaluate the agreement as well as the level of measurement error between devices.

Linear regression analysis and Pearson’s correlation coefficient (*r*) were performed to evaluate the linear relationship between paired measurements from the two devices. The linear equation derived from the curve fitting was plotted against a reference line which represents the perfect fit between measurements (i.e., with a slope coefficient equal to 1 and constant equal to zero). The best agreement of estimates is assumed when values of the slope coefficient and the constant of the fitted curve are close to those of the reference line and, hence, the two curves are overlapping. For the interpretation of the strength of correlation, the following cut-offs of Pearson’s *r* were adopted: trivial (<0.1), small (0.1–0.3), moderate (0.3–0.5), large (0.5–0.7), very large (0.7–0.9) and nearly perfect (>0.9) [32].

Intraclass correlation coefficient (ICC) was calculated with a two-way mixed-effects, absolute agreement, single rater/measurement model to evaluate the reliability between measurement devices. For interpretation, ICC values of less than 0.5, between 0.5 and 0.75, between 0.75 and 0.9, and greater than 0.90 were indicative of poor, moderate, good and excellent reliability, respectively [33]. ICC values are provided with 95% confidence interval and Cronbach’s alpha.

Standard error of the estimate (SEE) was used to evaluate the variation of measurements around the linear regression line. It was calculated as the standard deviation of the residuals of the regression model, with smaller values indicating a better estimation [6]. The standard error of measurement (SEM) and smallest detectable change (SDC) were computed to evaluate the error in terms of magnitude and sensitivity, respectively. The SEM is used as an indicator of absolute reliability [34] and was calculated as follows:(2)$$\mathrm{SEM}\;=\;\mathrm{SD}\;\sqrt{1-\mathrm{ICC}}$$
where SD and ICC represent the sample standard deviation and the reliability coefficient, respectively. The SDC is an indicator of the sensitivity of the measurement scale and is calculated from the SEM (1.96 × √2 × SEM). The obtained value represents the minimum change that needs to be detected to be confident that the observed change is real and not a result of the measurement error.

#### 2.4.2. Curve Similarity Analysis

To evaluate the similarity of velocity curves between the two measurement systems, two different analyses were performed. First, a cross-correlation analysis between paired time-series data was used to evaluate the spatial similarity of velocity curves [35]. This analysis produces the cross-correlation function (*R_xy_*), which is composed of negative or positive correlation coefficients (−1 < *r* < +1) at different phase shifts (*τ*) between the curves. The maximum value of the cross-correlation function *R_xy_*(*τ*) occurs at the specific phase shift where the two curves are most similar. Second, one-dimensional statistical parametric mapping (SPM) with multiple paired t-tests was used to objectively identify through statistical inference the curve regions which differ between the two measurement systems [36]. Briefly, this analysis evaluates the trajectory of the time-varying function *SPM*{*t*} with respect to a critical threshold. If the trajectory crosses the threshold at any time point, the null hypothesis is rejected and, hence, the curves are significantly different. The open-source spm1d package (version 0.4.8, https://www.spm1d.org, accessed on 26 October 2022) was implemented into the MATLAB^®^ software to perform the analysis on paired velocity curves. The other statistical tests were performed in Microsoft Office Excel^®^ (Microsoft Inc., Redmond, WA, USA) and SPSS v25 (SPSS Inc., Chicago, IL, USA).

## 3. Results

### 3.1. Discrete Velocity Parameters

The Bland–Altman plots of the event-specific velocity parameters are presented in Figure 3 (please refer also to Table 2 for mean difference and LoA values). The highest level of agreement between devices was observed at Vmax1 and PreStickV (mean difference ± standard deviation [95% LoA]: 0.00 ± 0.03 m·s^−1^ [−0.05, 0.06], and 0.02 ± 0.02 m·s^−1^ [−0.03, 0.06], respectively). A systematic bias of −0.04 m·s^−1^ was observed for Vmin (−0.04 ± 0.04 m·s^−1^ [−0.12, 0.03]), while the other parameters showed an average difference between the IMU- and video-based measurements of 0.00 ± 0.05 m·s^−1^. The limits of agreement averaged across the different parameters indicated that errors could go up to ~0.9 m·s^−1^, with the lowest level of agreement being found in EMV (−0.03 ± 0.06 m·s^−1^ [−0.14, 0.08]).

The results of linear regression and Pearson’s correlation analyses are shown in Figure 4. The correlation analysis showed large-to-very large associations between the paired measurements from the two devices for all parameters, with all correlation coefficients being greater than 0.7 (min-max *r* range: 0.70–0.91, *p* < 0.003; please refer to plots in Figure 4 for the specific *r* of each parameter). The analysis of the best-fit regression lines and their equations indicated that overall slope coefficients and constants had respective values greater than 0.7 and lower than 0.04. R^2^ values indicated that the level of goodness of fit varied across different parameters, mainly ranging from 0.55 to 0.83. The best and worst goodness of the linear fit were found in Vmax1 and EmV, respectively (R^2^: 0.83 vs. 0.50; *r*: 0.91 vs. 0.70; SEE: 0.02 vs. 0.05 m·s^−1^). A small variation around the regression lines was observed, as demonstrated by the SEE values ranging from 0.02 to 0.05 m·s^−1^ (please refer to plots in Figure 4 for the specific SEE of each parameter). The visual inspection of linear regression plots suggested that there is an over/underestimation of velocity above/below a specific threshold in some parameters (e.g., ~−0.45 m·s^−1^ in EMV, ~0.25 m·s^−1^ in CmV). However, the majority of the experimental data points are below those velocity thresholds.

The level of reliability between devices was assessed as good for most of the parameters (Cronbach’s alpha > 0.80, min-max ICC range: 0.69–0.90) (Table 2). Confidence intervals showed that there is a certain degree of variation in single-measurement ICC which can range from poor to excellent in specific parameters. The best reliability was found in Vmax1 (Cronbach’s alpha = 0.95, ICC: 0.95 [0.76–0.96]), while EmV and PreStickV parameters showed the lowest ICCs (0.69, moderate reliability).

Most of the parameters showed a low magnitude of measurement error, as indicated by the SEM values mostly ranging from 0.01 to 0.02 m·s^−1^ (min-max SEM range: 0.01–0.03 m·s^−1^) (Table 2). The SDC values indicated that the sensitivity of the measurement scale is overall below 0.07 m·s^−1^ (min-max SEM range: 0.02–0.07 m·s^−1^). The greatest measurement error with the lowest sensitivity was found in EmV, although the difference compared to the other parameters was on average 0.01 m·s^−1^.

### 3.2. Curve Similarity Analysis

Figure 5 shows the results from cross-correlation and SPM analyses for the evaluation of velocity curve similarity between devices. The analysis of cross-correlation functions of paired measurement indicated a high level of similarity between curves with minimum lag. The maximum value of the function averaged across trials was 0.95 ± 0.04 (mean ± SD) and occurred at a time lag of −1 ± 2% of the total task duration (Figure 5a). This indicates that the velocity curves obtained from the IMU device have a very small-to-no delay with respect to those from the video. The SPM identified two suprathreshold clusters, indicating that the velocity curves are significantly different in the ranges of 2–6% (*p* = 0.0056) and 95–100% (*p* = 0.0057) of total task duration (Figure 5c).

## 4. Discussion

The present study aimed to assess the validity of a single IMU-based methodology for the automatic estimation of barbell velocity profile and metrics at key time events of maximal Paralympic bench press. The set of statistical tests indicated a good level of agreement and large-to-very large correlation with the video-based reference device for most of the parameters. A moderate-to-good level of reliability and low magnitude of measurement errors were indicated by the ICC and SEM, respectively. Additionally, the proposed solution showed very high coherence of barbell velocity profile with reference device in terms of spatial similarity, with small differences at the very start and end of the exercise.

In their recent review, Clemente et al. summarized the level of validity and reliability of eight commercially available IMU models for barbell velocity assessment [10]. In the valid models, the reported level of error in velocity estimation (SEE) ranged from 0.01 to 0.18 m·s^−1^ and Pearson’s correlation coefficient ranged from 0.61 to 0.98. When considering only the reviewed studies that validated the use of IMUs in the bench press exercise, SEE and Pearson’s *r* ranged 0.135–0.18 m·s^−1^ and 0.76–0.97, respectively [5,6,37,38]. In this regard, our results demonstrated a lower error of velocity estimates (maximum SEE value was 0.05 m·s^−1^) and similar *r* values (all parameters had values greater than 0.70). The best results were observed in the Vmax1 (SEE = 0.02 m·s^−1^; R^2^ = 0.83; *r* = 0.91), the velocity metric that refers to the sticking point of barbell trajectory. Overcoming the loss of barbell velocity that occurs at this point is a critical factor for the successfulness of the lift at high, maximal or sub-maximal loads [15,39,40]. Therefore, information about the timing and velocity related to the sticking point would largely support athletes and coaches to develop specifically tailored training programs for performance improvement. To the authors’ knowledge, there is no current automatic and commercially available way to characterize the barbell velocity profile at the sticking point/region.

Grounded on clinical considerations and previous works, Courel-Ibañez and colleagues proposed benchmarks for the level of disagreement between reference and testing devices that can produce errors in load estimation (%1RM) during the bench press exercise [6]. For instance, differences between absolute mean velocity recordings from the two devices ranging 0.07–0.09 m·s^−1^ were suggested to indicate a moderate level of disagreement implying an error of about 5% in 1RM estimation. Based on this benchmark, the IMU-based event-specific metrics of barbell velocity present a good level of agreement with those obtained from the reference device. The mean difference between paired measurements from the two devices, averaged across all parameters, was 0.00 ± 0.04 m·s^−1^. However, it has to be noted that the benchmarks indicated by Courel-Ibañez and colleagues mainly refer to the measurement of mean velocity during the concentric phase [6]. If we consider the corresponding metric in the proposed method (i.e., CmV), both mean difference (0.00 ± 0.04) and 95% limits of agreement (−0.07, 0.07) were below that threshold of disagreement between devices. Therefore, these results suggest that the barbell velocity metrics obtained with the proposed methodology are within a good level of agreement with the video-based reference device, implying an error of less than 5% 1RM for load estimation.

The analysis of the similarity between kinematic time series is one of the modalities to evaluate sports performance [41,42] and movement in general [35,43,44]. In Paralympic powerlifting, assessing the similarity of barbell velocity profiles between dominant and non-dominant sides has been recently proposed as a tool to evaluate symmetry, which is a fundamental factor of good performance in competition [21]. However, the adopted methodology involved the use of video recordings, thereby requiring both space for optimal camera positioning and time for data processing. In this study, we attempted to reduce the costs related to this previous methodology by validating a standalone IMU-based solution to obtain the barbell velocity profiles. The cross-correlation analysis showed that the velocity profiles derived from the IMU had a very high spatial similarity with those extracted from video recordings. Diversely, the SPM analysis revealed that the signal differed at the very beginning of the eccentric phase and the end of the concentric phase, respectively at the first and last 5% of the total duration. The visual inspection of velocity curves suggested that these differences are related to a delay and anticipation of the negative and positive peaks of the velocity signal, respectively. Considering the results from [25], these differences are likely due to errors in the detection of event timings between the measurement systems. Specifically, a constant delay of 0.12 s (~3% of total movement duration) was found in the identification of the End event. This error was suggested to be linked to the different source signal and functions for event detection within the algorithm (i.e., maximum vertical marker displacement from video recordings vs. zero-crossing of linear acceleration from IMU). Despite that, the present results showed that barbell velocity profiles of the two measurement systems are of similar shape for the remaining 90% of total movement duration, where most of the important events of the bench press occur (i.e., Stop, Vmax1, and Vmin).

This study presents some limitations. To include all key events of Paralympic bench press, especially the sticking point (i.e., Vmax1), only maximal intensity was tested (~1RM). Consequently, it remains to be investigated whether the performance of the proposed solution would change at lower intensities. Another limitation is linked to IMU positioning. In this study, we were limited for practical reasons to place the sensor on the bar internally and close to the athlete’s handgrip, while the marker was positioned laterally on the barbell endcap. This different positioning may have produced changes in velocity estimation between the sensor and the reconstructed marker velocity that are linked to the deformation capacity of the barbel at high loads. Noteworthy, there is no consensus on barbell sensor positioning for bench press assessment as different locations were adopted in previous studies (e.g., athlete’s forearm [6,38,45,46], center [8] or right end [37,38] of the barbell). Therefore, further investigation on the effects of different sensor positioning should be performed to evaluate and eventually minimize this potential measurement error.

The approach proposed in this study discloses a novel way to assess barbell velocity through IMU. The combination of the automatic detection algorithm with IMU technology allows to immediately obtain more detailed discrete and continuous velocity metrics which can potentially unveil subtle changes in motor performance. Implementing this approach into the current IMU models available in the market will improve the number and quality of metrics provided to the user. This will uncover further practical applications, such as the evaluation of bilateral symmetry, that are now limited by the use of few discrete metrics or time-consuming data processing [21]. This approach has also the potential to be translated to other relevant strength exercises (e.g., squat and deadlift), although further studies are needed for ad-hoc adjustments of the algorithm to consider the exercise-specific velocity profiles.

## 5. Conclusions

The proposed IMU-based solution extends the number of available metrics to characterize the barbell velocity profile and monitor specific aspects of the maximal Paralympic bench press lifting sequence. Velocity parameters at key time instants were obtained automatically by implementing a custom-made algorithm and proved to be valid against a video-based reference device. Overall, measurement errors were below the values reported in previous literature and showed good reliability. Additionally, the IMU-derived barbell velocity profile showed very high similarity with the one obtained from the reference device, suggesting its use for further analysis, such as symmetry investigation. In conclusion, the proposed methodology represents a promising standalone solution capable of providing comprehensive and ready-to-use information with a low error of velocity estimate from which coaches and athletes can benefit for improving training protocols and sport performance in Paralympic powerlifting.

## Figures and Tables

**Figure 1 sensors-22-09904-f001:**
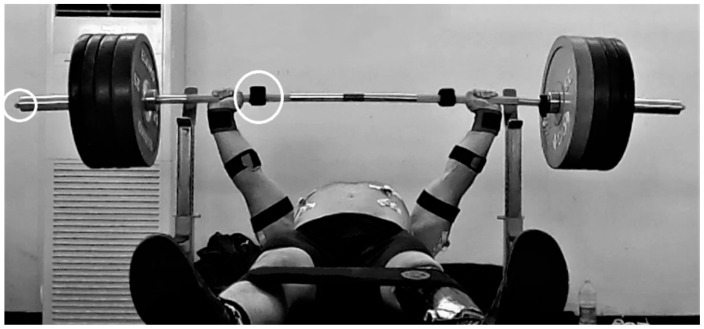
Picture of the experimental setup. White circles indicate the positioning of the marker and IMU on the barbell endcap (left circle) and bar (right circle), respectively.

**Figure 2 sensors-22-09904-f002:**
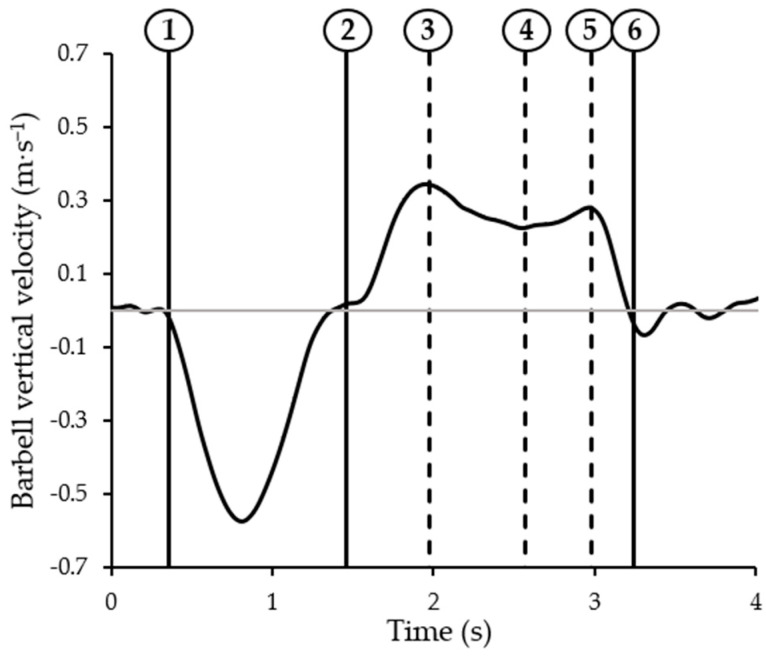
A typical example of barbell vertical velocity and events of the Paralympic bench press: Start (1), Stop (2), Vmax1 (3), Vmin (4), Vmax2 (5), and End (6).

**Figure 3 sensors-22-09904-f003:**
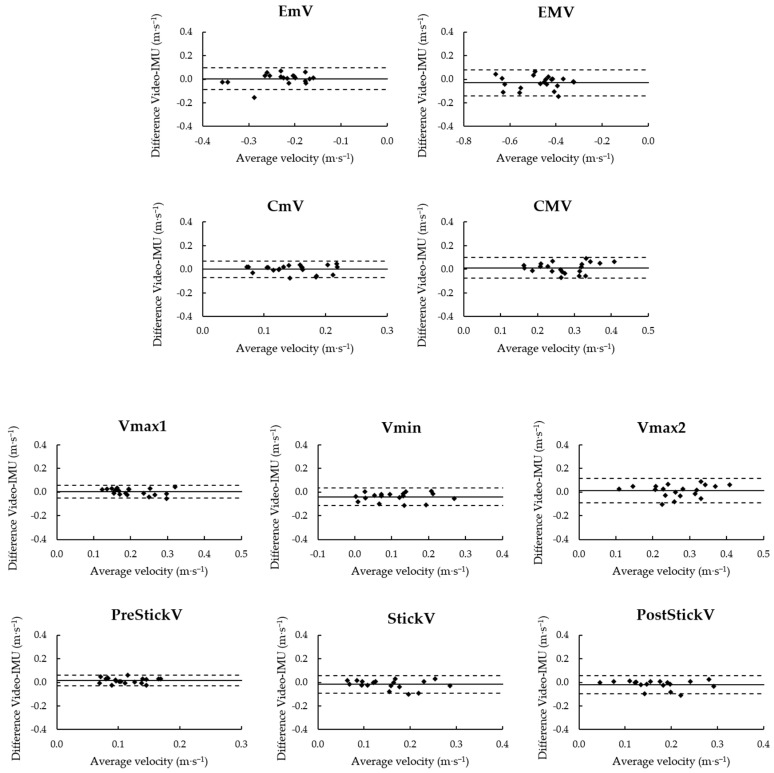
Bland–Altman plots of the event-specific velocity parameters. In each plot, the difference between video- and IMU-based measurement systems is displayed on the vertical axis and the average of paired velocity measurements from the two devices is on the horizontal axis. Horizontal solid and dotted lines indicate the mean difference and limits of agreement, respectively.

**Figure 4 sensors-22-09904-f004:**
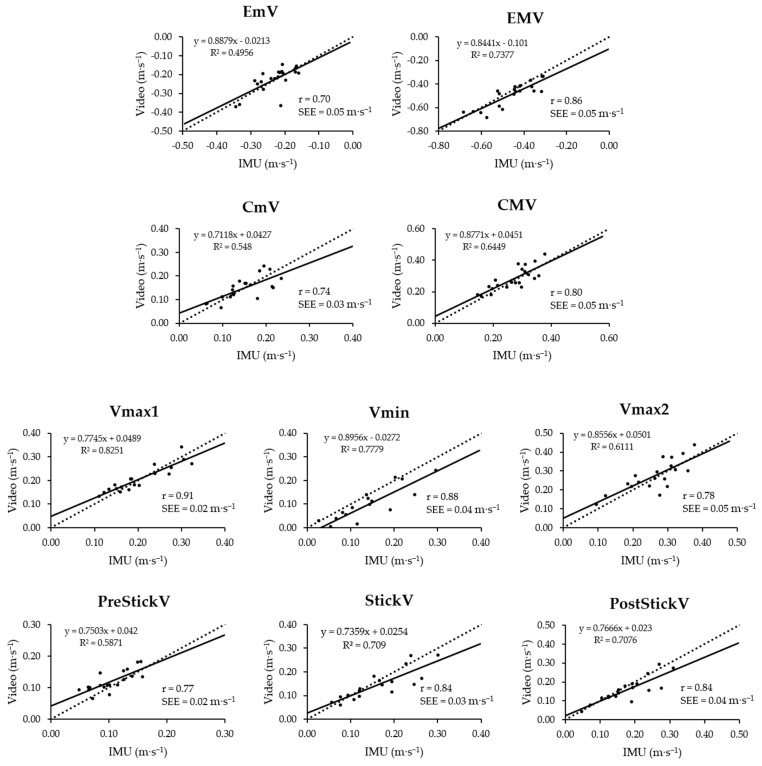
Results from the linear regression analysis between paired measurements from IMU and video devices. Linear regression equations (solid lines) are provided within each plot altogether with the perfect fit line (dotted lines), with video- and IMU-based measurements as independent (x, horizontal axis) and dependent (y, vertical axis) variables, respectively. R^2^, Pearson’s correlation coefficient (*r*) and standard error of estimate (SEE) values are also provided.

**Figure 5 sensors-22-09904-f005:**
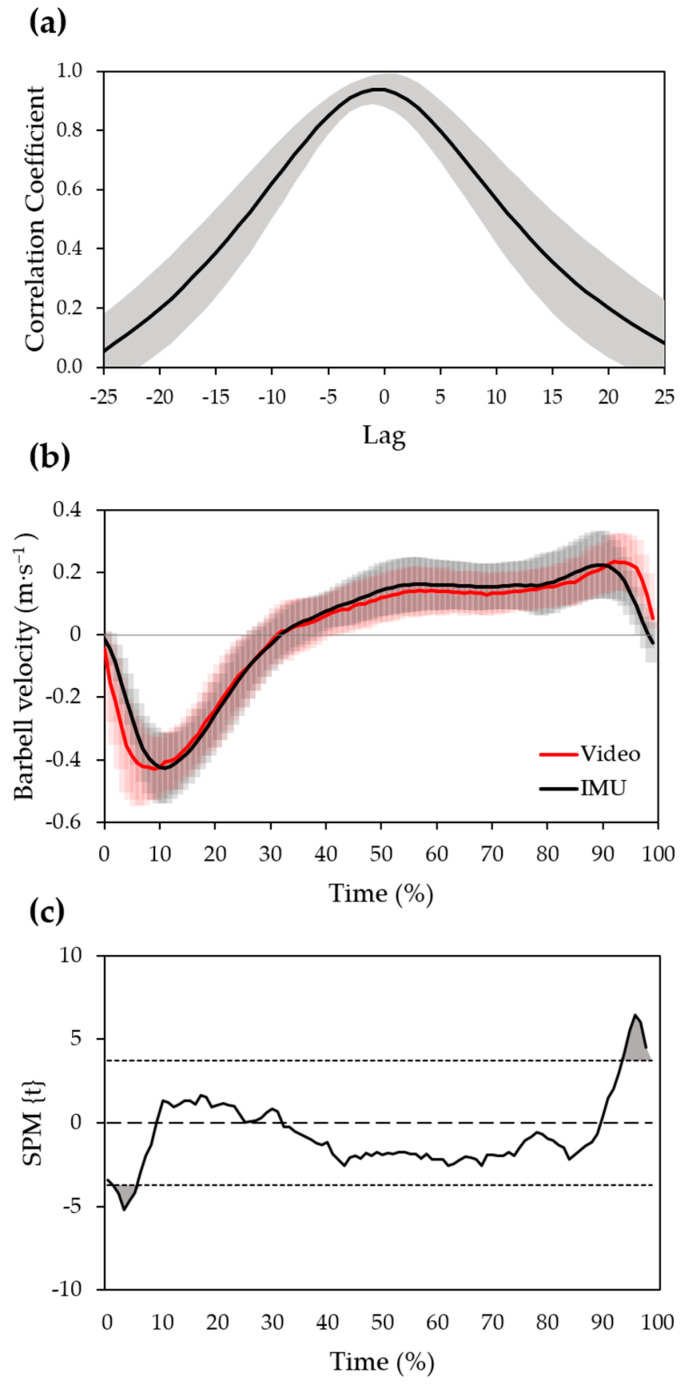
Graphical representations of the statistical analyses for the evaluation of velocity curve similarity between devices. (**a**) Ensemble average of cross-correlation functions plotted against 25% phase lag (positive and negative) with respect to the entire trial. Shaded area represents ± 1 SD. Positive peak coefficient indicates that velocity profiles from the two devices are in-phase, while positive/negative lag indicates that the signal derived from IMU device leads/follows the signal derived from the video. (**b**) Ensemble average of time-normalized barbell velocity profiles obtained from video (red) and IMU (black) devices. Shaded area represents ± 1 SD. (**c**) SPM{t} statistics as a function of time-normalized total task duration. The horizontal dotted lines indicate the critical threshold at |t| = 3.736 (α = 0.05) and the shaded areas at 2–6% and 95–100% ranges indicate the suprathreshold clusters.

**Table 1 sensors-22-09904-t001:** Demographic data, mean ± SD. RM: repetition maximum.

Age (Years)	30.5 ± 4.3
Sitting height (cm)	71.6 ± 6.8
Mass (kg)	72.5 ± 16.4
Impairment type (n. athlete)	- arthrogryposis (1) - spina bifida (2) - mono-lateral amputation (3) - bilateral amputation (1)
1RM	148.4 ± 38.6
Strength training experience (years)	12.8 ± 7.4
Powerlifting training experience (years)	5.4 ± 4.1
Powerlifting competition experience (years)	4.6 ± 4.0

**Table 2 sensors-22-09904-t002:** Outcomes of validation statistical analyses of discrete velocity parameters.

	Mean ± SD Video-IMU Difference (m·s^−1^)	95% LoA (m·s^−1^)	Cronbach’s Alpha	ICC [95% CI]	SEM (m·s^−1^)	SDC (m·s^−1^)
EmV	0.00 ± 0.05	[−0.09, 0.10]	0.81	0.69 [0.38–0.86] *	0.03	0.07
EMV	−0.03 ± 0.06	[−0.14, 0.08]	0.92	0.83 [0.59–0.93] *	0.02	0.06
CmV	0.00 ± 0.04	[−0.07, 0.07]	0.85	0.75 [0.47–0.89] *	0.02	0.05
CMV	0.01 ± 0.04	[−0.08, −0.10]	0.89	0.80 [0.57–0.91] *	0.02	0.06
Vmax1	0.00 ± 0.03	[−0.05, 0.06]	0.95	0.90 [0.76–0.96] *	0.01	0.02
Vmin	−0.04 ± 0.04	[−0.12, 0.03]	0.94	0.78 [0.12–0.93] *	0.02	0.05
Vmax2	0.01 ± 0.05	[−0.09, 0.12]	0.88	0.78 [0.52–0.91] *	0.02	0.07
PreStickV	0.02 ± 0.02	[−0.03, 0.06]	0.87	0.69 [0.25–0.88] *	0.01	0.03
StickV	−0.02 ± 0.04	[−0.09, 0.06]	0.91	0.81 [0.56–0.93] *	0.02	0.05
PostStickV	−0.02 ± 0.04	[−0.09, 0.06]	0.91	0.82 [0.55–0.93] *	0.02	0.05

SD: standard deviation; LoA: limits of agreement; ICC: intra-class correlation coefficient; CI: confidence interval; SEM: standard error of measurement; SDC: smallest detectable change. ** p* < 0.001.

## Data Availability

The data presented in this study are available on request from the corresponding author.

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
