# Peer review of "Validation of an Automatic Inertial Sensor-Based Methodology for Detailed Barbell Velocity Monitoring during Maximal Paralympic Bench Press"

_sensors, 2022, doi:10.3390/s22249904_

Round 1
Reviewer 1 Report
This paper proposes a new method to automatically obtain the complete barbell velocity curve and key performance indicators, which is novel and practical. But I think there are still a few things to improve.
Most of the articles show their research in words and diagrams, but there is no picture display in the human experiment part, such as Section 2.2, which makes it difficult to prove the authenticity of their work, and it is recommended to supplement.
The discussion part of Section 4 is too long, it is recommended to condense the language and description
I hope the author can discuss the improvement direction of this method in the future and other application scenarios, whether the weightlifting only for the bench press is too single.
Author Response
Please refer to the uploaded word document for answers to reviewers.

Reviewer 2 Report
1. Please state the novelty/ contribution of the research to knowledge clearly.
2. Please state the organization of the paper at the end of the introduction.
3. Line 133-3134, please state the unit of each of the parameters.
Author Response

(The authors gave the same response as above.)

Reviewer 3 Report
First of all, I wanted to congratulate the authors for the great work done. I just have a few questions to resolve.
Introduction
In the introduction, paragraph 1, the authors indicate that the use of commercial devices is very common among users. Please provide more information on these devices (if they have also been used in scientific research, not only commercially, if the ones normally used are validated, etc.).
Moreover, as the objective is currently stated, the novelty of the article is not well understood. An algorithm has been determined in previous research, so what does yours contribute? A new device that includes this algorithm? Or what is really the novelty? Please specify this, as well as research hypothesis.
Methods
Please provide more information on warm up. How long did it last, how many sets, reps, %RM, speed? All this information would be necessary to be able to replicate the study.
The statistical analysis is well described.
Results
The results section is well described and presented.
Discussion
In the second paragraph of the discussion the authors indicate "and similar r values (9 out of 10 parameters had values greater than 0.74)." They then describe the moment with the best r value. But what about the moment with the worst r value? What is it and what could it be due to? Please justify this.
A paragraph on the practical applications derived from the present research would be needed following the limitations of the study.
Author Response

(The authors gave the same response as above.)

Round 2
Reviewer 1 Report
The authors have completed the revision of the paper. All the contradictory parts and mistakes that I suggested or warned in the previous version have been corrected or expressed clearly in the last revision. I think this paper can be published.
Author Response
We thank the Reviewer for the time and effort spent revising our manuscript.